Second Reply to the Reviewer's comment for Article #11: Comparison of Physical Distances Between Pedestrians on a Street in the Central Area of Osaka City Before and After the Covid-19 Pandemic Based on Deep Learning Techniques

Atsushi Takizawa, Haruka Narumoto

Shinpei Ito, Nagahiro Yoshida

We appreciate the time and effort you have dedicated to providing insightful feedback on strengthening our article. Thus, it is with great pleasure that we resubmit the article for further consideration. We have incorporated changes that reflect the detailed suggestions you have graciously provided. We also hope that our edits and the responses we provide below satisfy the issue and concern you have noted. The revised article is marked in red font.

## Reply to Reviewer

**Review comment**

> There is a very small typo, please check it and correct it.
>
> Section 3.2
>
> A real-time kinematic global n avigation satellite system (RTK-GNSS)
>
> ->
>
> A real-time kinematic global navigation satellite system (RTK-GNSS)

**Author response**

We fixed the typo miss.

**Article revision**

| P4 | A real-time kinematic global navigation satellite system (RTK-GNSS) |
|----|---------------------------------------------------------------------|

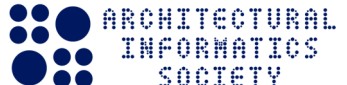

# Comparison of Physical Distances Between Pedestrians on a Street in the Central Area of Osaka City Before and After the Covid-19 Pandemic Based on Deep Learning Techniques

Atsushi Takizawa*[1], Haruka Narumoto*[2], Shinpei Ito*[3] and Nagahiro Yoshida*[4]

[1] Professor, Graduate School of Human Life and Ecology, Osaka Metropolitan University
[2] Bachelor, Department of Human Life Science, Osaka City University
[3] Title, Nikken Sekkei Research
[4] Associate Professor, Graduate School of Engineering, Osaka Metropolitan University
* takizawa@omu.ac.jp

## Abstract

COVID-19 has been spreading worldwide since 2020. Although the World Health Organization recommends maintaining a physical distance of at least 1 m among people, the Japanese government recommends 2 m. In this study, we used deep learning and other techniques to statistically compare and verify the change in the physical distance between pedestrians on a sidewalk in a large Japanese city before and after the COVID-19 pandemic. A video-based approach was used to accomplish this. The video before the COVID-19 pandemic was recorded in October 2018 in the Namba area of Midosuji, Osaka City. For comparison, new videos were recorded at the same location in October 2020. YOLOv3 SPP was applied to automatically extract a large number of pedestrians on the street. Three observation areas were set on the sidewalk within the target area, and the physical distances between the pedestrians were measured. Two indices were used to measure the physical distance: the average nearest neighbor and Ripley's K-function. Thus, the change in the physical distance between people on the street, before and after the COVID-19 pandemic, could be quantitatively and statistically compared. The results showed an increase in physical distance after the COVID-19 pandemic, which depended on the state of behavior, density, and human relations.

## Keywords

COVID-19, Physical distancing, Pedestrian detection, Osaka, Yolo, DeepSort, Average nearest neighbor, Ripley's K-function

**Type**: Research article

**Citation**: F. Author et al. "How to write a peer-reviewed paper of the Journal of Architectural Informatics Society: ver. 20210329". Journal of Architectural Informatics Society, vol. 0, no. 0, pp. a1-aXX. doi: https://doi.org/xx.xxxx/xxxx/xxxxx

**Received**: 15 April 2020
**Revised**: 29 December 2021
**Accepted**: 05 January 2021
**Published**: 10 January 2021

## 1. Introduction

COVID-19 has been spreading worldwide since 2020, and the pandemic has shown no signs of ending in 2022. In Japan, preventive measures such as wearing masks, frequent hand washing, and elimination of dense crowds have been advocated since the beginning of the pandemic, although there are some differences in infection control measures adopted among countries [1]. The first two are synonymous with ensuring physical distancing, as is the case in other countries. Physical distancing is defined as maintaining a certain distance between people to prevent droplet infection; for example, the World Health Organization (WHO)

recommends maintaining a distance of at least 1 m [2], and the Japanese government recommends maintaining a distance of at least 2 m [3]. Although vaccines and medicines for the virus have been developed, maintaining physical distance is still considered an important and basic infection control measure. For example, Kato et al. [4] estimated from cell phone company data that the chance of residents to go out to the city center was reduced by half in the suburbs of Osaka Prefecture immediately after the outbreak of the COVID-19. Wellenius et al. [5] estimated the impact of the declaration of a state of emergency in the United States on changes in human flow at the state level using Google mobility data. These studies analyzed changes in human behavior on a macro spatial scale. Meanwhile, there are few studies that quantitatively measure the degree to which people have accepted the maintenance of physical distance before and after the pandemic.

In this study, we used deep learning and other techniques to statistically compare and verify the change in physical distance between pedestrians on a sidewalk in Namba which is a downtown shopping district in Osaka City before and after the COVID-19 pandemic. The video before the COVID-19 pandemic used in this study was recorded in October 2018 from a building across the street near the Namba intersection. It was a part of an experiment [6] to measure and verify the effectiveness of sidewalk expansion in the Namba area of Midosuji street. For comparison, new videos were captured at the same location in October 2020, and the proposed deep learning technique was applied to automatically extract a large number of pedestrians on the street.

The novelty and academic contributions of this study are as follows. First, as noted above, such studies themselves are rare. The conditions of a suitable location for measuring outdoor physical distance during the COVID-19 pandemic include the following: the area is sufficiently pandemic for the COVID-19, the area is overcrowded to the extent that physical distance becomes a problem for people, and there are many strangers present in the area. In light of these conditions, Osaka Prefecture has the second highest number of PCR-positive cases for COVID-19 in Japan [7] after Tokyo, and Namba is one of the busiest areas in Japan. The target area is a sidewalk where an unspecified number of people pass by. The target area of this study is therefore suitable for the purpose of the study. To the best of the authors' knowledge, data taken before and after the COVID-19 pandemic at the same season and target area, which is suitable for measuring such physical distance, are extremely valuable. On the other hand, it should be noted that this study targets only this area, which limits the generality of the study.

Secondly, the technical contribution is that while using a relatively new deep learning method as a means of measuring pedestrians, the system incorporates several improvements to increase its practicality, and evaluates the accuracy of detection and distance between pedestrians. Because the measurement of pedestrians is a time-consuming task, technologies to automate it have been developed. These techniques are summarized in the next section; however, in this study, we used a video-based approach [8]. In the proposed method, the target pedestrians are extracted from still images captured at specific frame intervals. To measure the spatial distribution of pedestrians, the video-based approach offers a better balance between cost and accuracy than other methods, although privacy must be considered when using the method in a small space where faces can be distinguished. In the past, pedestrians were manually marked. However, with the development of image processing technology, deep learning techniques for detecting objects such as people have been developed and used in recent years. Faster R-CNN [9] and YOLO [10] are typical examples of such techniques. In this study, we used YOLO as a pedestrian detector. Because the images were captured at an angle from the top of the sidewalk, pedestrians often overlapped in them. We used a dataset called CloudHuman dataset [11], which is a collection of crowd images, and images of the target area, to re-train YOLO from scratch. In order to project the detected pedestrians to planar coordinates, we proposed a method that can estimate parameters stably even with many corresponding points. And we evaluated the accuracy of pedestrian detection and distances between pedestrians.

Finally, we devised methods for measuring and testing physical distance. We set three observation areas on the sidewalk within the target area and measured the physical distances between pedestrians. Generally, the distance to the nearest pedestrian is considered to be a measure of physical distance, but this alone may not be sufficient because the physical distances within a close group are expected to be smaller. Therefore, considering the grouping nature of pedestrians, their tendency to aggregate was also evaluated as a physical distance using the K-function, which is used in spatial statistics [12]. Thus, the changes in the physical distances between people on the street, before and after the COVID-19 pandemic, could be quantitatively and statistically compared, and some of the effects of the pandemic on the activities of pedestrians in the outdoor space of the city center can be understood.

This study deals with a very simple problem: the change in the physical distance between

pedestrians on a sidewalk. However, such basic study is time-consuming when done manually and has rarely been conducted. We show the fact that deep learning techniques have made it possible to obtain pedestrian data with high spatial resolution and reasonable reliability. This is a major step forward in architectural and urban planning. We hope that building on this fundamental knowledge and technology, with architectural informatics playing a central role, will lead to a deeper understanding of space and planning.

The remainder of the paper is organized as follows. Section 2 reviews the current literature. Section 3 describes the preparation of videos of the target area. Section 4 details the estimation of pedestrian position through detection, tracking, and projective transformation. Section 5 describes the methods employed for measuring physical distancing. Section 6 reports the overall change in physical distance. Section 7 provides a detailed analysis of physical distance. Section 8 contains a discussion on some issues, and Section 9 concludes the paper.

## 2. Literature review

When surveying pedestrian data, it is necessary to estimate the position of pedestrians with high accuracy over a long period of time. The measurement of pedestrian positions has been widely used in the fields of architectural planning, urban planning, and traffic engineering to understand their usage of various facilities and spaces. For example, counting cross-sectional traffic volumes, determining the number of people and their movement trajectories from videos [8], and stalking surveys of pedestrians following their routes [13] have been used for a long time. However, these surveys require considerable time and cost for implementation. For example, if the area to be observed is large or the density of pedestrians is high, a large number of personnel and measurement devices must be deployed to perform these surveys. Furthermore, when using videos, the number and locations of pedestrians must be manually obtained from each frame, which also requires a significant amount of labor.

There are several methods for identifying and detecting pedestrians, and many practical methods to do so automatically, such as Wi-Fi [14], Bluetooth [15], and LIDAR [16], which can measure the distance to the target, primarily indoors. Although Wi-Fi and Bluetooth require devices such as a smartphone and an access point, they are widespread and inexpensive; therefore, they can measure human flow over a wide area at a relatively low cost. However, the resolution of these methods depends on the distance between access points, and they do not provide the spatial resolution at least in ten centimeters, which is used in this study. In contrast, a distance sensor such as a laser can measure the distance to an object with millimeter or centimeter accuracy, but using such sensors is generally expensive and costly for measuring an area. Global positioning systems (GPSs) are mainly used outdoors [17], and although high-precision GPS is used for surveying, it is common to use the location information from pedestrian smartphones to monitor human flows, and various location data are already available [18], [19]. However, the accuracy of GPS installed in smartphones is not high, especially in urban areas where buildings are densely located, and errors of tens of meters or more are common. Additionally, the installation of a specific application on a smartphone is imperative to continuously acquire logs to determine the location, which is a problem that hinders comprehensiveness. Compared with these pedestrian detection methods, the method using video images can provide a better balance between cost and accuracy. The idea of detecting pedestrians from videos existed prior to deep learning becoming popular [20], and various techniques to accomplish this have been developed owing to advancements in image processing technologies [21]. With the recent development of deep learning, its development has been rapid and practical usage has been demonstrated. Pedestrian detection includes tasks such as object detection, semantic segmentation, instance segmentation, and pose estimation. The most common task among these is object detection, wherein the detection target is surrounded by a rectangular bounding box. Object detection methods can be roughly classified into faster R-CNN and YOLO types. Faster R-CNN is expected to have a high accuracy, but the estimation speed is slow; in contrast, YOLO is slightly less accurate but has a faster detection speed. In semantic segmentation, the detection region is applied to the shape of the category to be detected, rather than a bounding box, and the image is segmented into regions. One of the most popular semantic image segmentation methods is DeepLab v3+ [22]. Instance segmentation is an extension of semantic segmentation, wherein segmentation is performed by distinguishing different objects within each category. Although these segmentation tasks are more detailed than object detection, they require significant computational speed and have difficulty in detecting the entire body of a shadowed pedestrian when there is a pedestrian overlap. The last task, pose estimation, represents a person as a network model of the skeleton and estimates the pose. Typical algorithms for this

include OpenPose [23]. In this study, the bounding box method is suitable because only the central position of the pedestrian on the plane needs to be identified. Moreover, information on the approximate position and height of the pedestrian is required, even if they are overlapped. Therefore, we used YOLOv3, a typical object detection method, for pedestrian detection. Examples of pedestrian detection using YOLO are in [24], [25], [26].

This study tracked two specific pedestrians for an accuracy evaluation, as described in Section 4. To improve the efficiency of this task, we used a task called tracking, which continuously tracks the same object between different frames. The tracking task can be divided into single object tracking, which tracks a single object, and multi-object tracking (MOT), which tracks multiple objects simultaneously. An MOT task was employed in this study. Old tracking methods use conspicuous markers to track objects [27]; however, with the development of deep learning, methods to perform tracking without any special equipment have been proposed. DeepSORT [28] is a popular method for tracking-by detection; however, it is still in its infancy, and the technology is still evolving.

Finally, several studies have used object detection methods to measure physical distances after the outbreak of COVID-19 [29], [30], [31], [32]. Compared with these studies, this study uses YOLO trained on a pedestrian dataset that considers crowd overlap and local conditions to improve detection accuracy, compares the same locations before and after the outbreak of COVID-19, and uses Ripley's K-function and the distance to the nearest person as the physical distance. Ripley's K-function, which is a spatial statistics method, was used to statistically evaluate the spatial distribution, in addition to the physical distance to the nearest person.

## 3. Preparing videos of the target area

This section explains the video recording in the target area.

### 3.1 Target area and video recording periods

The target area of the video recording was the area on the southeast side of the Namba intersection, which is on the north side of the Namba area. Namba is the second largest shopping district in Osaka City (see Figure 1). The sidewalk is along Midosuji street, which is a typical street in Osaka City. To make Midosuji a space where people can gather in 2025 during the Osaka Expo, a policy was implemented in 2018 to convert two lanes of side roads into sidewalks, out of the current six lanes of roadways [33]. As a result, the widening work of the sidewalk was planned and executed from the Namba station, on the south side of Midosuji, to the north. To verify the effectiveness of sidewalks, social experiments were carried out. Kawachi and Yoshida [6] recorded videos of a social experiment in October 2017, and conducted research on the behavior modification of pedestrians due to the widening of sidewalks. This study used videos taken in October 2018 from the eighth floor of a building on the southwest side of Midosuji Street. Figure 2 shows the target area to be recorded from the video camera angle and Figure 3 shows the ground plan of the target area.

The COVID-19 pandemic began in Japan in spring of 2020. We took another video of the sidewalk from the same location in October 2020 to match the timing of the 2018 video. Figure 4 shows the changes in the daily number of people infected with COVID-19 in Osaka Prefecture in 2020. October 2020 was between the second and third waves of COVID-positive cases. However, this number did not drop to zero, and there were approximately 50 people infected per day in the prefecture.

However, between 2018 and 2020, there was a slight difference in the pedestrian pool in front of the pedestrian crossing on the northern side. As of October 2018, the roadway in this area has been converted into a sidewalk. As shown by the red oval in Figure 5, a fence was built on the west side of the north-side crosswalk in 2020. The widths the of pedestrian crossing and pool were wider in 2018. When comparing the physical distances in Section 5, it is necessary to note this difference in spatial situations.

### 3.2 Video recording and surveying

The dates, times, and weather conditions of the video recordings are listed in Table 1. The videos were recorded using a Sony Handycam FDR-AX700. Because it was necessary to record the video through the glass from the inside of the building, a polarization filter was installed to reduce the glass reflection. The scale of the plan view (Figure 3) of the target area was determined through a survey conducted on October 14, 2020. A real-time kinematic global navigation satellite system (RTK-GNSS) survey was conducted using RWP by Biz Station Co. Ltd. The RTK reference station was JP-RJBE 10, registered at www.rtk2go.com

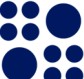

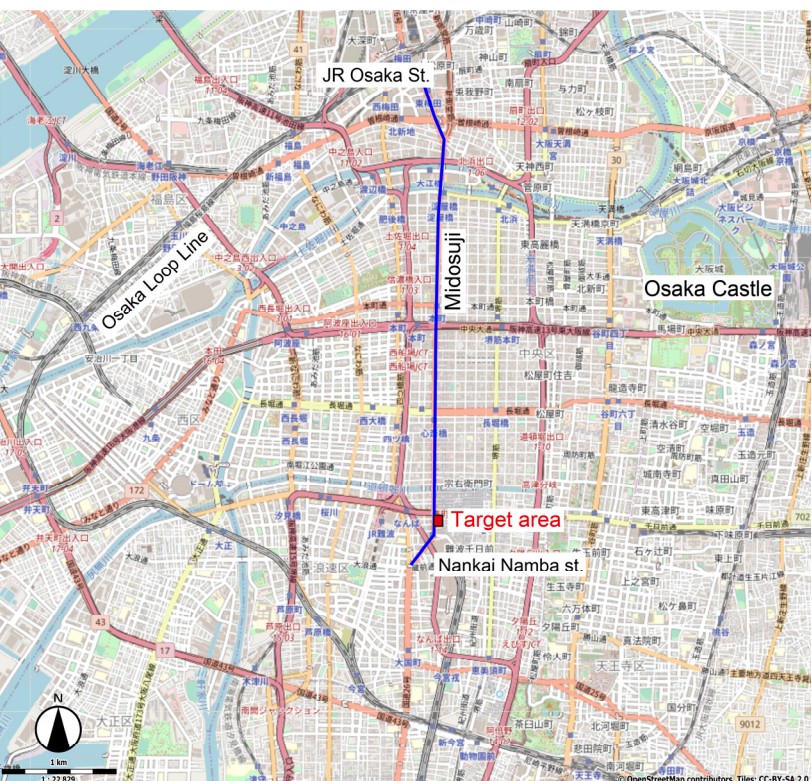

Figure 1. Target area in Osaka City

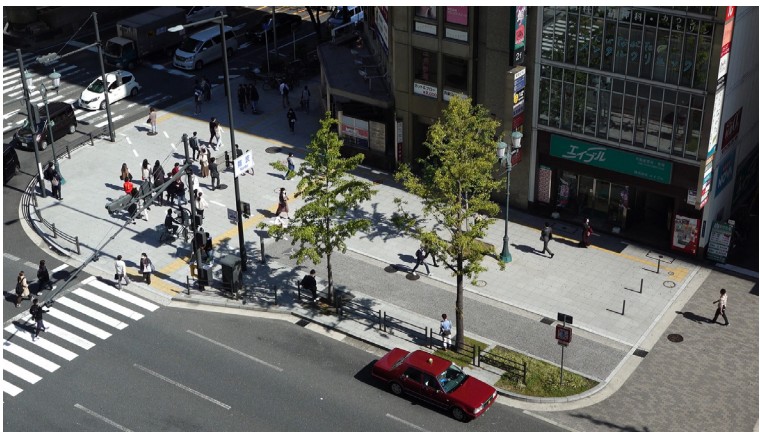

Figure 2. Target area to be recorded

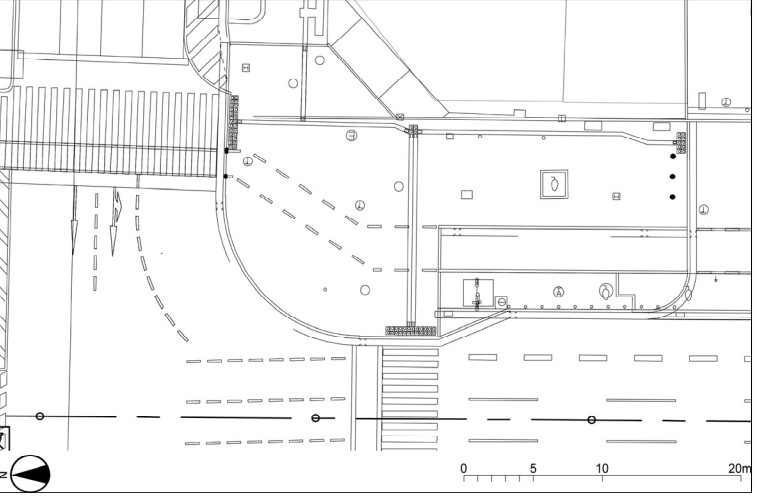

Figure 3. Ground plan of the target area

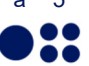

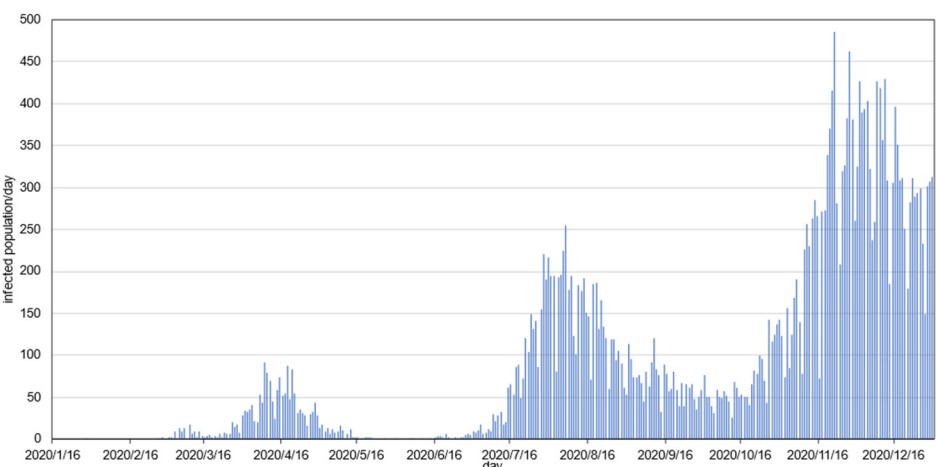

Figure 4. Changes in the number of COVID-19 cases per day in Osaka Prefecture in 2020 [31]

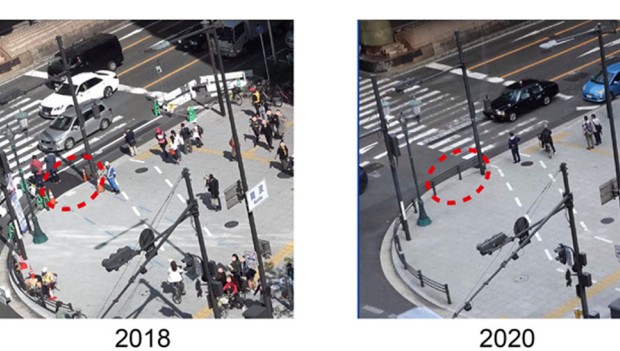

Figure 5. Changes in the northern part of the target area

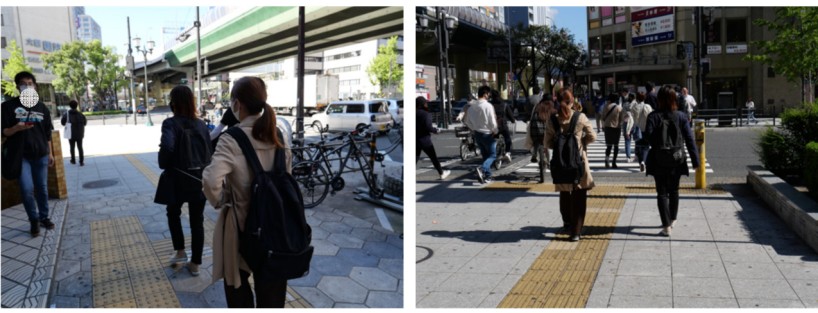

(a) Vertical  (b) Parallel

Figure 6. Measurement for accuracy evaluation of physical distance

Table 1. Parameters of the video recordings

| Date, year | Time | Weather at 12PM | Temperature (Max/Min, °C) |
|---|---|---|---|
| Oct. 13, 2018 (Sat.) | 11:50~20:00 | Cloudy | 24.3/13.2 |
| Oct. 15, 2018 (Mon.) | 11:50~20:00 | Sunny | 23.4/17.6 |
| Oct. 14, 2020 (Wed.) | 11:30~17:00 | Sunny | 25.8/16.6 |
| Oct. 24, 2020 (Sat.) | 11:30~17:00 | Sunny | 19.8/13.8 |

in Higashinada Ward of Kobe City. The error was 1.5 cm when the RTK survey was fixed. The longitudes and latitudes of the six feature points on the ground were also surveyed. The coordinates were input to a geographic information system and corresponded to the feature points of the plan image with an image size of 3,220 × 1,920 pixels. Consequently, one pixel of the plan image was estimated to be approximately 0.01687 m. To evaluate the accuracy of the physical distance in Section 5, two participants walked either vertically or in parallel from 11:30 to 11:55 on October 24, 2020, in the east-west and north-south directions in the target area while maintaining spacing with a 1 m string (see Figure 6). The measurement was conducted on a public street where many pedestrians were passing by, and if the string was

firmly fixed to the participants, there was a risk of an accident if another pedestrian tried to pass between the two participants. Therefore, we did not fix the string to the body, but only held it in the hand with the center of the body or the center of the side of the body in mind, so that the string could be released in case of an emergency. This scene was also recorded on video at the same location.

## 4. Estimating pedestrian position through detection, tracking, and projective transformation

In this section, we describe pedestrian detection and tracking. The pedestrian detection framework used in this study is as follows. A general object-detection technique based on deep learning was used to estimate the bounding box of each pedestrian for the still image in each frame extracted from the video. Next, the representative points of the bounding boxes were transformed into plane coordinates by the projection transformation method, and this was used for the measuring the distance of the pedestrian However, when measuring the distance between two pedestrians for accuracy evaluation, to facilitate the identification of the person to be measured, pedestrian tracking is performed after object detection. Each process is described in the following subsections.

### 4.1 Extracting still images from videos

Because there were several interruptions of approximately 1 min each in the videos of all dates due to battery replacement, the still images were cropped to 1,920 × 1,080 pixels at 1 fps from the 30-minute video of each time frame of [12:15–12:45, 13:15–13:45, 14:15–14:45, 15:15–15:45], which were shot continuously for the 4 days listed in Table 1, and saved as a jpg image. The number of still images per 30 min was 1,800; therefore, 7,200 images were extracted for each day. The video for evaluating the accuracy of the physical distance was edited into a movie of 4 min that only contained the clips of the two participants shown in Figure 6, and 2,400 images were similarly cropped at 10 fps.

### 4.2 Pedestrian detection

The YOLOv3 SPP [34] was used for object tracking. A pre-learned model, learned using a standard MS COCO 2007 dataset [35], can be downloaded from the site in [36]. The MS COCO dataset includes annotation data for 80 object classes, wherein more than 60,000 images include a person class. Additionally, although the number of people per image, including the person class, is approximately four on average, the number of pedestrians in street images required for city planning surveys is generally larger. Therefore, when pedestrians on streets are detected by a model learned by MS COCO, there remains a concern about its learning accuracy. Additionally, in general object detection datasets such as MS COCO, the bounding box of a person is enclosed only in the visible area of the person. Therefore, for example, in the case of an image wherein only the upper body is visible, the position of the person is estimated to be deeper relative to the camera, than the actual position when the image is transformed into plane coordinates by the projection transformation.

Furthermore, a dataset called CrowdHuman (CH), which collects images of dense crowds, was released in 2019. The CH dataset contains 15,000, 4,370, and 5,000 images for training, validation, and testing, respectively. There are 23 persons per image, with various types of occlusions in the images in the dataset. Each human instance is annotated with a head-bounding box, human visible-region bounding-box, and human full-body bounding-box. In this study, we use the CH dataset of the full-body bounding box to learn YOLO from scratch with the expectation of improving the detection accuracy even when pedestrians overlap, considering the position of the plane. However, as shown in Figure 2, there are two street trees in the images of the target area, which are expected to reduce the detection accuracy because they block the view of pedestrians passing behind them. Therefore, we randomly sampled 50 still images of the target area on a holiday from the 2018 image set and annotated each pedestrian in each image with a rectangle around the entire body, including invisible areas. We actively annotated pedestrians hidden behind trees that could be recognized by the human eye. After preparing the dataset, we trained YOLO. First, it was trained with 15,000 images from the CH dataset. Next, 250 training images consisting of randomly sampled 225 images from the CH dataset and 25 images of the target Midosuji Street (CHM) were used for additional learning of the previously learned model.

In the network settings of YOLO to be trained, the number of classes was changed from 80, which was based on the MS COCO dataset, to one because only persons were detected, and

the filter size of the convolutional layer affected by this modification was reduced from 255 to 18. The hyperparameters of YOLO for learning are as follows, wherein the underlined parameters are those that were changed from the defaults: batch = 64, subdivisions = 32, width = 608, height = 608, channels = 3, momentum = 0.9, decay = 0.0005, angle = 0, saturation = 1.5, exposure = 1.5, hue = 0.1, learning_rate = 0.001, burn_in = 1,000, max_batches = 50,000 (CH) or 60,000 (CHM), policy = steps, steps = 40,000, 45,000 (CH) or 55,000, 58,000 (CHM), and scales = 0.1, 0.1.

The trained YOLO detector was evaluated through mean average precision (mAP) using the remaining 25 test images of Midosuji Street. Table 2 lists the mAPs of the three YOLO detectors trained with MS COCO, CH, and CH and CHM. Detectors trained with CH alone often detected pedestrians incorrectly when they were hidden by trees or in the shadows at the back of the image; however, additional learning with CHM significantly improved the detection accuracy in such areas. Therefore, in the present study, we used a detector trained with CHM for pedestrian detection to obtain the highest mAP[*1].

Finally, we detected pedestrians in 7,200 images from each date and 2,400 images were used for the accuracy evaluation of the distance by using the YOLO learned with CH and CHM.

Table 2. mAPs of detectors for 25 test images for each YOLO with different training datasets

| MS COCO | CH | CH and CHM |
|---|---|---|
| 0.633 | 0.670 | 0.876 |

## 4.3 Pedestrian tracking

We evaluate the accuracy of pedestrian spacing in Section 5. As explained in section 3.2, the two participants with a cord must be identified in each frame. To streamline this task, DeepSORT technique is applied to identify the same pedestrians in each frame. It combines the Kalman filter and person re-identification technique, and performs tracking by matching the image in the bounding box detected as a person through any detection method between adjacent frames. Judging whether a person represented in different images is the same is called person re-identification, for which we used the DG-Net method [37]. DeepSORT tracks objects in real-time depending on the implementation; however, we implemented offline tracking. The flow of our offline pedestrian tracking system using DeepSORT is as follows. First, the bounding box area estimated as a person from the image of each frame is extracted by YOLO, and the image of each area is prepared. DG-Net is applied to these images and 512-dimensional feature vectors are obtained. Then, a CSV file is generated wherein the positions and feature vectors of the bounding boxes in the image are arranged in the order of pedestrians in each frame. Finally, based on this information, the same ID is assigned to the person's image that DeepSORT estimates as the same person.

Figure 7 shows an example of tracking performed on an image for accuracy evaluation. Tracking was reasonably stable when there was little overlap of obstructions or pedestrians; however, in other cases, there were often detection omissions and ID changes. Therefore, the final determination of the participant was performed by checking the tracking results.

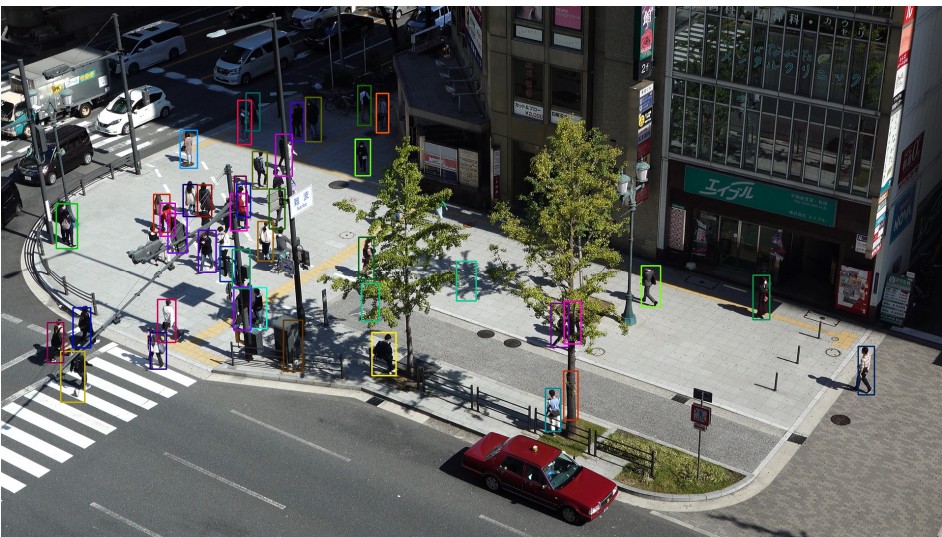

Figure 7. Tracking of images for evaluating the accuracy of physical distances

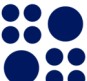

### 4.4 Projective transformation

Because each video is taken from an oblique angle, the coordinates of each pedestrian in the still image must be converted to planar coordinates through a projective transformation. To do this, the parameters of the projective transformation must be first determined. Let $P$ denote the set of corresponding points that show the same position in the still image and plan view, as shown in Figure 8. The coordinates of the pixel in the still image and those of the plan view at each corresponding point $p \in P$ are denoted as $pos_p = (x_p, y_p)$ and $pos'_p = (x'_p, y'_p)$, respectively. The projective transformation $f$ from $pos_p$ to $pos'_p$ is given as follows:

$$(x'_p, y'_p) = f(x_p, y_p) = \left( \frac{a_1 x_p + b_1 y_p + c_1}{a_0 x_p + b_0 y_p + 1}, \frac{a_2 x_p + b_2 y_p + c_2}{a_0 x_p + b_0 y_p + 1} \right). \tag{1}$$

In (1), $a_0$, $a_1$, $a_2$, $b_0$, $b_1$, $b_2$, $c_1$, and $c_2$ are the parameters to be calibrated, and we denote the parameter set as $A$. Because there are eight parameters, the values of these parameters can be obtained by solving a simultaneous linear equation if the coordinates of four different corresponding points are available. However, when the number of corresponding points is small, local conversion errors can be large depending on their locations. Therefore, as shown in Figure 8, we set nine corresponding points around locations where pedestrians tended to walk for each video recording day, and instead of solving the simultaneous equations, we numerically obtained the parameter values that minimized the conversion error in (1).

Under a certain parameter set $PRM$, let $ppos'_p = (px'_p, py'_p)$ denote the coordinates of the corresponding point estimated using (1). Note that in the context of our estimation method, $pos'_p = (x'_p, y'_p)$ are the real coordinates of the corresponding points in the plan view, and in general, $ppos'_p \neq pos'_p$. The problem of determining the values of these parameters can be formulated as the following quadratic programming problem:

$$\underset{PRM}{Minimize}\ error_q = \sum_{p \in P} \left( \left( x'_p - px'_p \right)^2 + \left( y'_p - py'_p \right)^2 \right). \tag{2}$$

In this study, (2) was solved using the nonlinear optimization function of the Excel solver; however, it depends on the initial value. If the initial value is not properly set, the error increases. Therefore, before solving this problem, an exact solution of the following linear programming problem that minimizes the sum of the absolute values of the transformation errors similarly derived from (1) was obtained using the Excel solver and set as the initial value for solving (2).

$$\underset{PRM}{Minimize}\ error_l = \sum_{p \in P} \left| x'_p \left( a_0 x_p + b_0 y_p + 1 \right) - \left( a_1 x_p + b_1 y_p + c_1 \right) \right| +$$
$$\sum_{p \in P} \left| y'_p \left( a_0 x_p + b_0 y_p + 1 \right) - \left( a_2 x_p + b_2 y_p + c_2 \right) \right|. \tag{3}$$

After optimizing the parameters for each video recording day, the mean value of $error_q$ is approximately 915. This is converted into root mean squared error (RMSE) per corresponding point to $\sqrt{915/9} \approx 10$ pixels. Because one pixel is 0.01687 m, as described in Section 2.2, the mean error for each corresponding point through the projective transformation is estimated to be approximately 0.17 m.

Using the projective transformation obtained above, the coordinates of the pedestrians detected in the image on the plane were obtained by considering the center of the bottom edge of the bounding box as the position of each pedestrian, as shown in Figure 9. According to the Japanese basic data for architectural design [38], a human body standing, viewed from above, is encompassed by a rectangle with a width of 50 cm and a height of 30 cm. If the center of this rectangle is regarded as the actual pedestrian center, the pedestrian center position based on the present calculation method will be shifted about 15 to 25 cm toward the camera from the actual center position. However, the purpose of this study is not to obtain the absolute coordinates of pedestrians, but to measure the distance between pedestrians. Locally, pedestrians tend to face the same or opposite direction because the target area is on the sidewalk. In particular, $w_2$ and $w_3$ might be dominated by pedestrians moving in the east-west and north-south directions, respectively. In $w_1$, there is a slight mixture of movement in both axes, but it is expected that more pedestrians will turn to the north-south when they stand. In any case, since the orientation of the pedestrians is considered to be generally aligned within each observation area, the center position of the pedestrians as a whole is

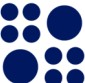

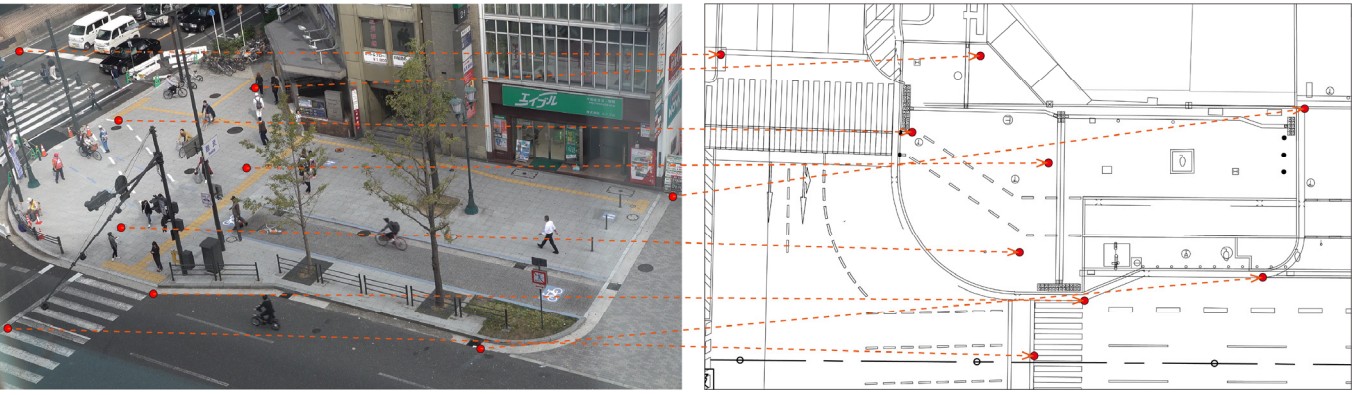

(a) Video image of the target area             (b) Plan view

Figure 8. Example of corresponding points for projective transformation (weekdays 2018)

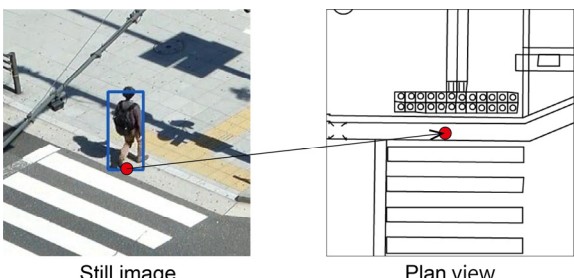

Still image          Plan view

Figure 9. Bounding box (left) of a pedestrian detected by YOLO and its position on the plan (right)

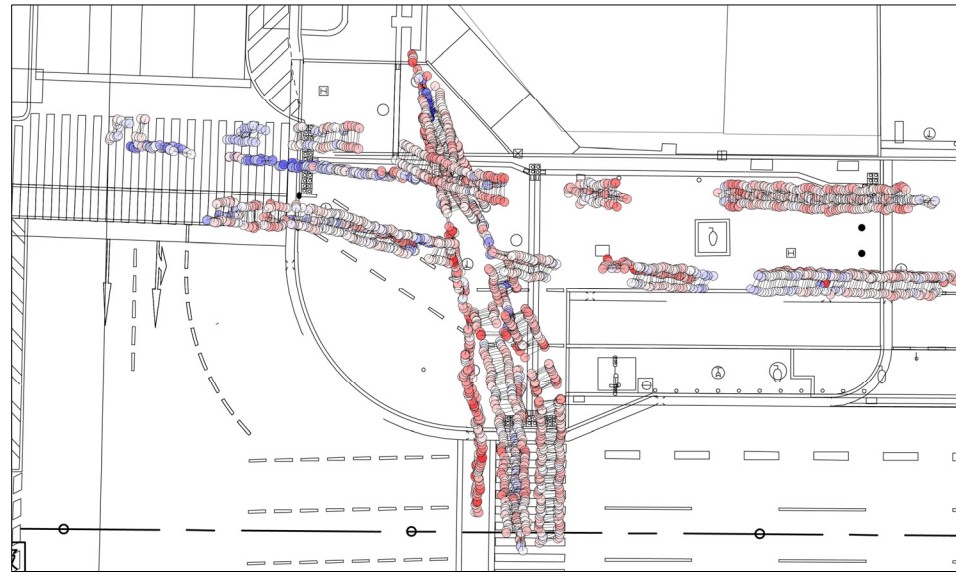

Figure 10. Visualization of the trajectory and physical distance between the two participants (blue < 1 m = white < red)

shifted parallel, and the effect on the measurement of the pedestrian spacing is not considered to be significant.

## 5. Accuracy evaluation of the interval between two pedestrians

The number of images wherein both participants were detected was 1,514 out of 2,400 still images, as described in Section 4.1. The positions of the two participants on the plane were obtained through projective transformation. Subsequently, by multiplying the Euclidean distance between them, the physical distance was estimated. Figure 10 shows a visualization of the trajectory and physical distance between the two participants. Pedestrians were detected within the camera's field of view, but the trajectory was interrupted by the shadows

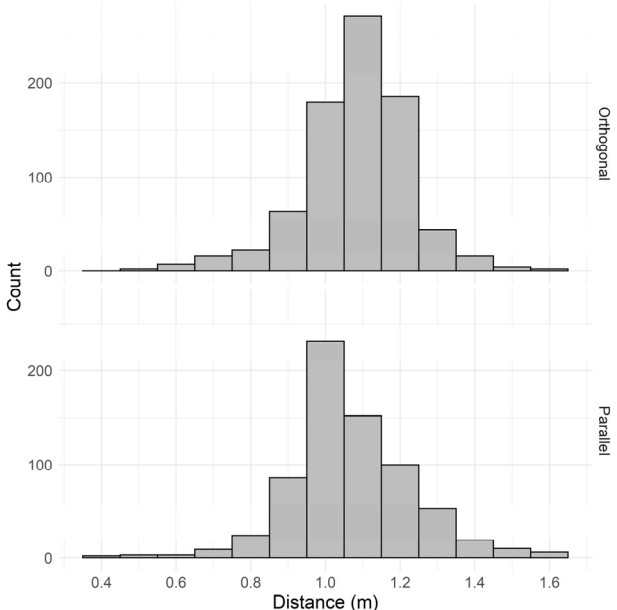

Figure 11. Histograms of distances between two participants

Table 3. Simple statistics of the distance between two participants (unit is meter)

|  | Mean | Standard Deviation (SD) | Interquartile Range (IQR) | 0% | 25% | 50% (Median) | 75% | 100% | N |
|---|---|---|---|---|---|---|---|---|---|
| All cases | 1.077 | 0.154 | 0.181 | 0.377 | 0.987 | 1.073 | 1.168 | 1.645 | 1,514 |
| Orthogonal | 1.084 | 0.143 | 0.160 | 0.500 | 1.011 | 1.097 | 1.171 | 1.643 | 816 |
| Parallel | 1.068 | 0.166 | 0.187 | 0.377 | 0.971 | 1.048 | 1.158 | 1.645 | 698 |

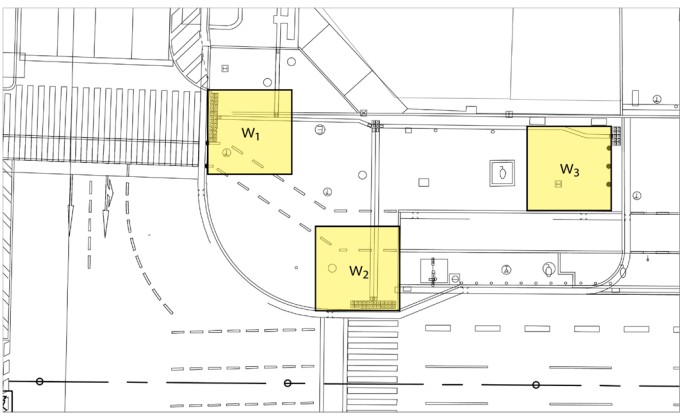

Figure 12. Measurement areas for physical distance between pedestrians

of trees. The upper-left corner of the plane view tends to slightly underestimate the distance. Histograms of the physical distance between the two participants are shown in Figure 11. Three types of histograms were created: one wherein the rows of participants are roughly "orthogonal" to the direction of the video camera, another wherein they are "parallel," and a third wherein no distinction is made (all cases). From these histograms, it is confirmed that there is little difference depending on the direction of the video camera and that the measured distances are normally distributed with a mean physical distance of approximately 1 m. Table 3 summarizes the physical distances between the two participants. The mean values are approximately 1.08 m, which is approximately 8 cm more than assumed. The relative error is maintained at approximately 8%, and evaluation of the physical distance of the pedestrian based on object detection and project transformation seems appropriate.

## 6. Measurement methods of physical distancing

First, the observation area of the physical distance is described, followed by the physical distance indices.

### 6.1 Setting the measurement area

As shown in Figure 12, square observation areas of the same size (approximately 400 pixels per side = 6.75 m) were set at three locations. The width of each square corresponds to the width through which the pedestrians pass. The three areas are represented by $w_1$, $w_2$, and $w_3$. The number of pedestrians entering the measurement area is represented by $n (\geq 2)$, and the Euclidean distance between pedestrians $i, j \in \{1, \dots, n\}, i \neq j$ is represented by $d_{ij}$. We assumed that the physical distance between pedestrians shows different aspects of walking and standing. $w_1$ and $w_2$ are areas of human pools in front of the pedestrian crossing, and stagnation and walking actions are periodically switched. By contrast, $w_3$ is an area that pedestrians always pass. As mentioned, the width of the pedestrian crossing and the presence or absence of fences adjacent to $w_1$ differed between 2018 and 2020. The passable boundary was narrowed by 2020. It is important to note the impact of this difference on the physical distance between pedestrians.

### 6.2 Average nearest neighbor (ANN)

The first indicator measures the distance to the nearest pedestrian in the measurement area and averages its value for each pedestrian. Formally, ANN is expressed by the following equation:

$$ANN = \frac{1}{n} \sum_{i=1}^{n} min\big(d_{ij} | j \in \{1, \dots, n\}, i \neq j\big). \tag{4}$$

### 6.3 Ripley's K-function

According to Hall's concept of personal space [39], personal distance can be divided into, in order of proximity, intimate distance (~0.45 m), personal distance (0.45–1.2 m), social distance (1.2–3.5 m), and public distance (~3.5 m). In terms of human relationships, the distances can be considered to represent family or lovers, friends or colleagues, acquaintances, and strangers, in order from closest to farthest. Thus, when pedestrians are walking as a group, they are likely to be closer, and the physical distance between them is expected to be smaller. Because the ANN is a local indicator of physical distancing, the intended physical distance may not be evaluated in such cases. Therefore, using Ripley's K-function developed in the field of spatial statistics, the pedestrian distribution in the observation area could be grasped in a global manner. Let $r (> 0)$ denote any radius from each pedestrian, $A (> 0)$ denotes the area of the observation area, and $p_{ij} (> 0)$ denotes the presence or absence of a weighted pedestrian. The value of the K-function at any radius $r$ is given by the following equation:

$$K(r) = \frac{A}{n(n-1)} \sum_{i=1}^{n} \sum_{j=1, i \neq j}^{n} I\big(d_{ij}\big),$$
$$\text{where } I\big(d_{ij}\big) = \begin{cases} p_{ij}, & d_{ij} \leq r \\ 0, & d_{ij} > r \end{cases}. \tag{5}$$

There exists a bias in the K-function, wherein the number of other points around a point closer to the boundary of the observation area is counted as less than the actual number. To address this bias, the weight was changed by introducing boundary processing. As described further in the paper, because the observation area in this study was set considering the boundary with walls, crosswalks, etc., special boundary processing was not conducted. That is, $p_{ij} = 1$ for any $i, j \in \{1, \dots, n\}, i \neq j$. $K(r)$ is the mean ratio value of the number of other points where the radius from each point in the observation area exists in $r$, multiplied by area $A$, which is a constant; however, when comparing for our purpose, it is better to know the ratio rather than the absolute value. Additionally, because the maximum value becomes uniform at 1 and is easy to understand, the value of $k(r) = \frac{K(r)}{A}$, which is not multiplied by $A$, is used in this study.

A curve of the K-function is plotted by increasing the radius $r$ gradually, repeatedly calculating the K-function, and plotting the value at that time. In this study, we draw a curve by increasing $r \in \{0.0, 0.1, \dots, 8.0\}$(m) at 0.1 m steps. As stated, the K-function represents the percentage of points; therefore, the curve shows the range of points that are easy or difficult to gather. By comparing the graphs in the case of Figure 13, we can estimate the tendency of spatial distribution, such as relatively concentrated, random, and uniform types,

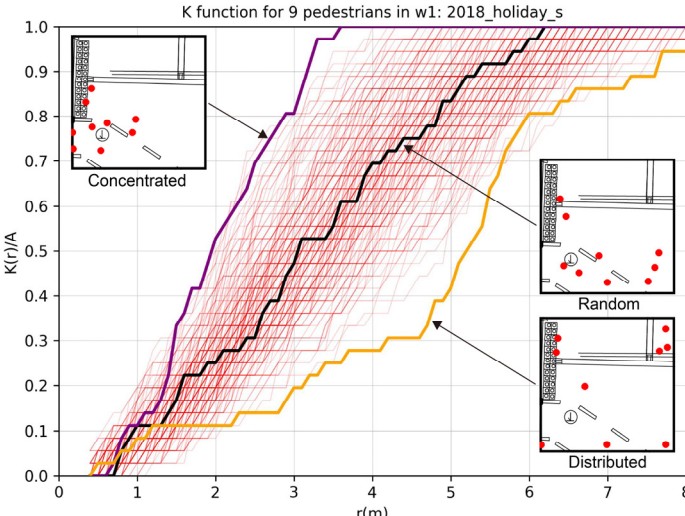

Figure 13. Relationship between curves of K-function and pedestrian distributions

from the value of the K-function, which is larger. It is also possible to estimate the tendency of spatial distribution such as relatively concentrated type, random type, uniform type, etc., from the graph with a larger value of K-function.

## 7. Overall change of physical distance between pedestrians

This section explains the measurement and comparison of physical distance between pedestrians, which is the primary subject of this study. The physical distance is measured by the ANN and K-function from the coordinates of the pedestrians on the plane obtained by pedestrian detection and transformation in each frame.

Table 4 lists the sum of the number of pedestrians detected in each observation area, and the mean number of pedestrians per frame. The number of pedestrians observed at $w_1$ was the largest. In the recording period of the data, maximum holidays occurred in 2018, and weekdays were the least in 2020. There was a difference of over three times in the numbers between these periods.

Table 4. Sum of the number of pedestrians detected in each observation area and the mean number of pedestrians per frame

| Observed area | Date | No. of pedestrians | Mean no. of pedestrians per image |
|---|---|---|---|
| $w_1$ | 2018_weekday | 51,395 | 7.1 |
| | 2018_holiday | 83,683 | 11.6 |
| | 2020_weekday | 24,106 | 3.3 |
| | 2020_holiday | 55,486 | 7.7 |
| $w_2$ | 2018_weekday | 38,491 | 5.3 |
| | 2018_holiday | 56,743 | 7.9 |
| | 2020_weekday | 18,522 | 2.6 |
| | 2020_holiday | 32,024 | 4.4 |
| $w_3$ | 2018_weekday | 20,053 | 2.8 |
| | 2018_holiday | 27,853 | 3.9 |
| | 2020_weekday | 8,939 | 1.2 |
| | 2020_holiday | 18,724 | 2.6 |
| Total | 2018_weekday | 109,939 | 15.3 |
| | 2018_holiday | 168,279 | 23.4 |
| | 2020_weekday | 51,567 | 7.2 |
| | 2020_holiday | 106,234 | 14.8 |

Figure 14 shows the overlaps of the spatial distribution of pedestrians detected in each of the 7,200 images. The darker the red color, the greater number of pedestrians were present at the site during the observation period. Dark red is noticeable in the observation areas $w_1$ and $w_2$.

| Year | Weekday | Holiday |
|------|---------|---------|
| 2018 | | |
| 2020 | | |

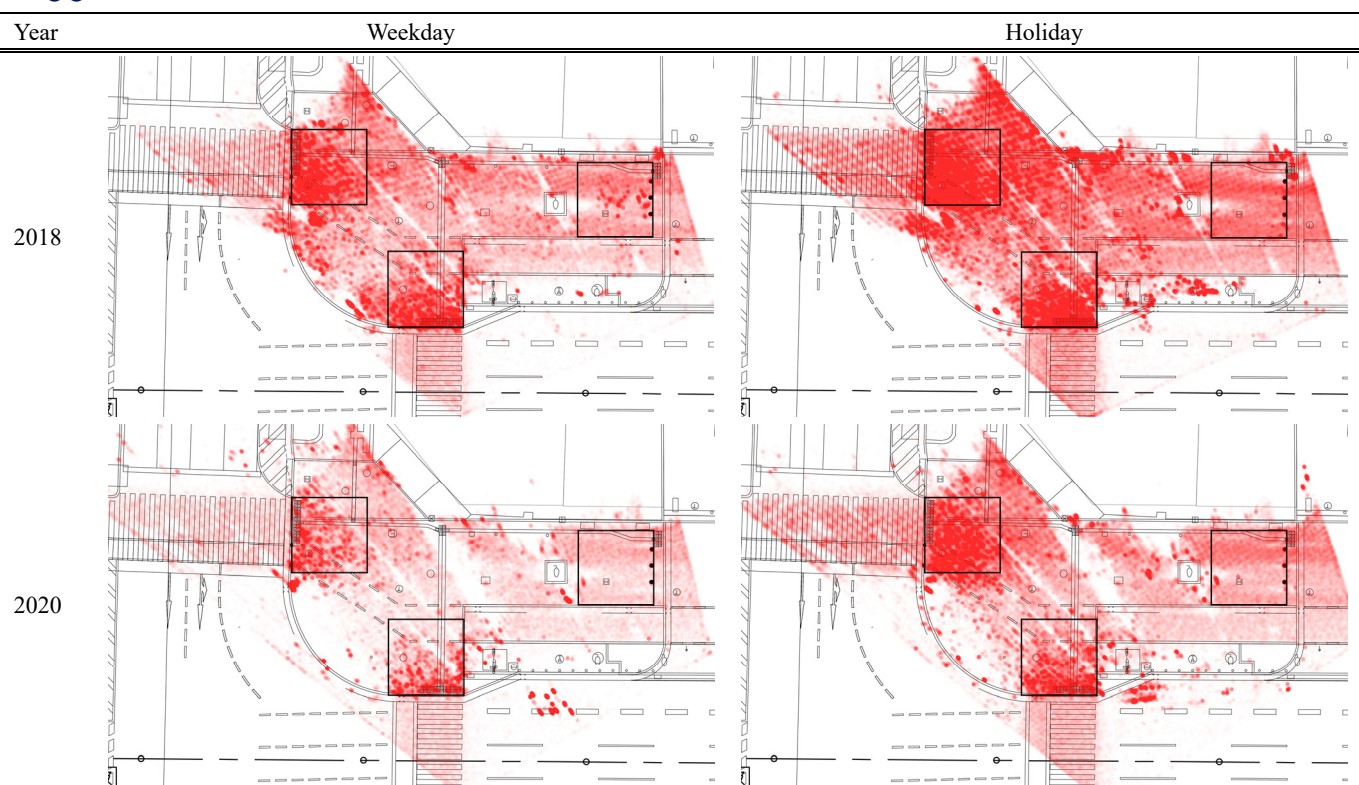

Figure 14. Cumulative distribution of pedestrians detected in 7,200 images for each date

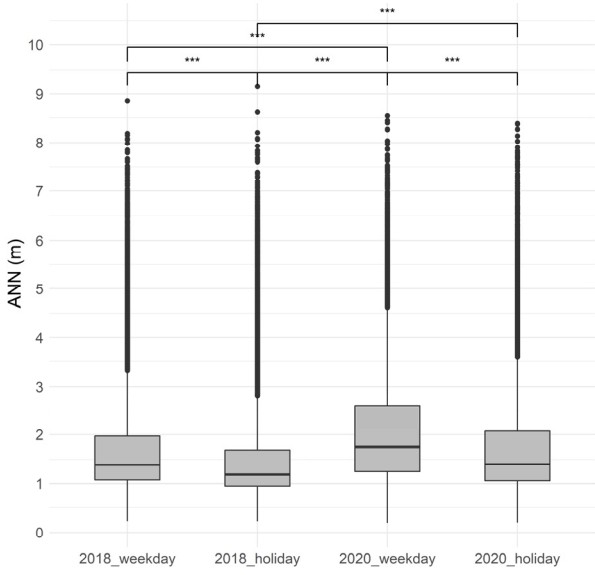

Figure 15. Boxplots of the ANN for each date and the result of multiple comparison (*** is 0.01 significance level)

Table 5. Basic statistics of ANN for each date combined with all observation areas (the unit is meter)

| Date | Mean | SD | IQR | 0% | 25% | 50% (Median) | 75% | 100% | N |
|------|------|------|------|------|------|--------------|------|------|------|
| 2018_weekday | 1.75 | 1.12 | 0.90 | 0.23 | 1.07 | 1.39 | 1.97 | 8.86 | 17,047 |
| 2018_holiday | 1.53 | 1.00 | 0.74 | 0.23 | 0.95 | 1.19 | 1.69 | 9.15 | 18,976 |
| 2020_weekday | 2.14 | 1.34 | 1.35 | 0.20 | 1.25 | 1.75 | 2.60 | 8.54 | 11,458 |
| 2020_holiday | 1.76 | 1.10 | 1.01 | 0.20 | 1.06 | 1.40 | 2.07 | 8.39 | 16,684 |

Figure 15 shows boxplots of ANN for each image set combined with all observation areas, and Table 5 lists their basic statistics. Because the frequency distribution of the physical distances tends to be asymmetric in either image set, we confirmed the tendency of ANN through the median value, and not through the mean value which assumes the normal

distribution. Multiple comparisons were performed using the Steel-Dwass test. Considerable differences were observed at significant level 0.01 in the distribution of ANN for each pair of dates, excluding weekdays in 2018 and holidays in 2020. Comparing ANNs for weekdays and holidays in 2018 and 2020, respectively, ANN after the COVID-19 pandemic was significantly larger than before, with median differences of 0.36 m and 0.21 m, respectively. However, even in the data for the weekday of 2020, when the most distance was available, the median ANN is 1.75 m, which does not reach the recommended value of 2 m.

## 8. Detailed analysis of physical distance

In this section, we analyze and compare the physical distances for each measurement area in detail. Only still images from when the traffic light was red, and pedestrians were standing, were used for measuring the physical distance between pedestrians at the observation areas $w_1$ and $w_2$. This is because, as mentioned in Section 5.1, it is assumed that the physical distance is different between walking and standing. As discussed below, this is a reasonable assumption because the physical distance does not change noticeably before and after the pandemic in $w_3$, wherein walking is predominant.

### 8.1 Comparison method of physical distance indices

Generally, there are predominantly businessmen on weekdays, and shopping and leisure pedestrians on holidays in the city center, and there appears to be a difference in the attributes of pedestrians based on the day of the week. Considering this, we compared the differences in physical distance between weekdays and holidays in 2018/2020 with the same number of pedestrians in the same observation area. The indices of the physical distance were calculated for the pedestrian distribution in each image. In the case of the ANN, one value was obtained for each image, and in the case of the K-function, one curve was obtained. Because multiple values and curves can be obtained from the image dataset of each year, it is necessary to perform a statistical test to compare the physical distance in different years.

Figure 16 shows a scatter plot wherein the results of the mean nearest neighbor distances of the corresponding images are superimposed. With a lower number of pedestrians, the variance of the distance increases, and as the number of pedestrians increases, the variance decreases. Simultaneously, the number of relevant samples also decreases. In this figure, two line graphs are drawn by connecting the mean and median values of distances for the same number of pedestrians. When the number of pedestrians is small, there is a difference between the mean and median values, and it is assumed that the distance distribution becomes asymmetric. Figure 17 shows a similar illustration for the K-function. However, the horizontal axis becomes the radius r to be searched, and the dimension to be considered to obtain the same condition is increased by one, in addition to the number of pedestrians in this case. In the case of the K-function, the available values are discrete. When the number of pedestrians is low, the number of value types is small, and a rattling curve such as Figure 17(a) is formed. However, when the number of pedestrians is large, the curve becomes smooth, as shown in Figure 17(b). The mean and median values of the K-functions for samples with the same radius were calculated, and their curves were generated.

Therefore, in this study, the difference in the distribution of the value of any index between video recording years was examined by the Mann–Whitney U test, which is a non-parametric test, and not by the t-test that assumes a normal distribution. Specifically, the set of datasets is denoted by $D = \{2018\_weekday, 2018\_holiday, 2020\_weekday, 2020\_holiday\}$, the set of observation areas is denoted by $W = \{w_1, w_2, w_3\}$, and the set of number of pedestrians observed is denoted by $P = \{2, ..., P_{max}\}$, where $P_{max}$ is the maximum number observed. Then, for all $d \in D$ and $w \in W$, let $ANN_d^w(p)$ denote the set of ANNs for a given pedestrian number $p \in P$. Furthermore, let $K_d^w(r, p)$ denote the set of values $K(r)$ at a certain distance $r \in R$. Finally, for two different datasets $d_1, d_2 \in D$, the null hypothesis for each index is set as follows, and the U test is conducted with a significance level of 0.05:

$$H_{ANN}: \text{The distribution of } ANN_{d_1}^w(p) \text{ and } ANN_{d_2}^w(p) \text{ is similar.}$$
$$H_K: \text{The distribution of } K_{d_1}^w(r, p) \text{ and } K_{d_2}^w(r, p) \text{ is similar.}$$

### 8.2 Results of ANN

Figures 18 and 19 show the comparison between the median ANNs of 2018 and 2020 for each observed area on weekdays and holidays, respectively. There was a significant difference in the distribution of the ANN in each year in the intervals with the colored

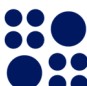

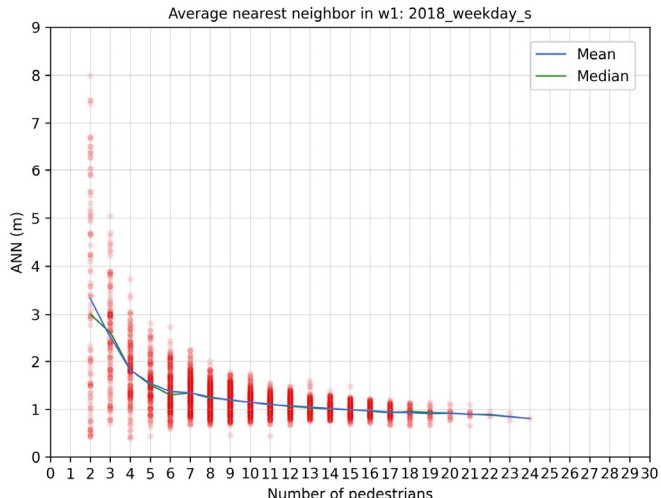

Figure 16. Scatter plot of ANNs for images with two or more pedestrians and line graphs of the mean and median for each number of pedestrians standing in $w_1$ on the holiday 2018

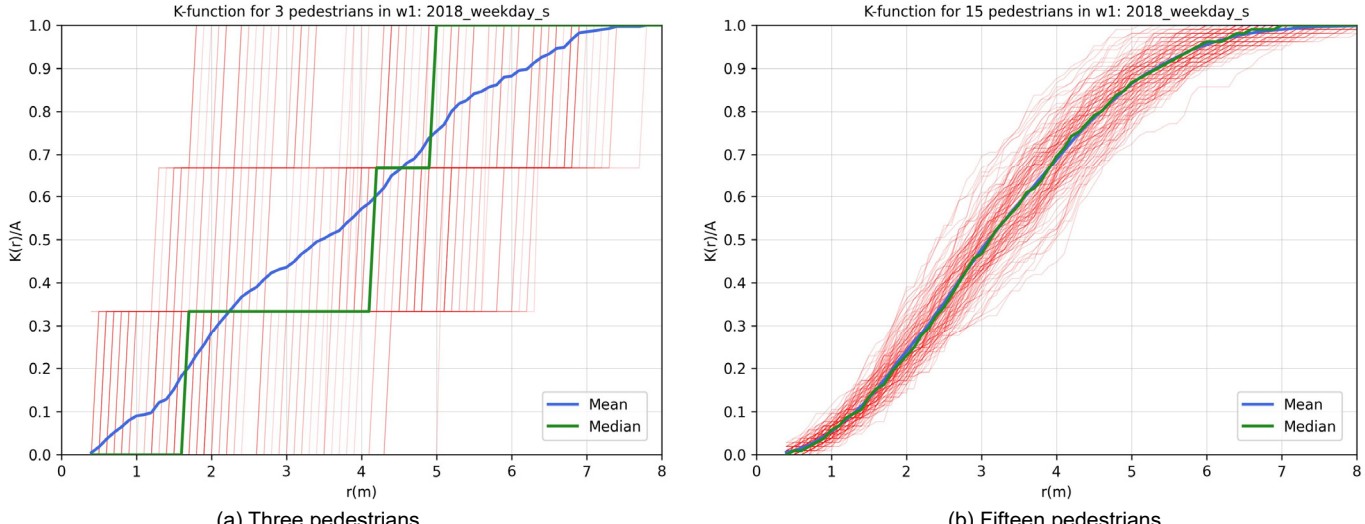

(a) Three pedestrians          (b) Fifteen pedestrians

Figure 17. Superposition of the red curves of each K-function in all images for three and 15 pedestrians, and the curves of their mean and median for each radius r in area $w_1$ on the holiday in 2018

backgrounds. Statistically larger intervals are shown as red and blue rectangles for the 2020 and 2018 data, respectively. First, the data on weekdays were confirmed. In the area $w_2$, when the number of pedestrians is five or less, the ANN is significantly larger in 2020, and when the number of pedestrians is three or less, the difference expands to 0.8 m, and the ANN over 2 m is maintained. However, the distance in this area is reversed by the number of pedestrians, and it is difficult to comprehend the tendency. Finally, although region $w_3$ was a non-stagnant site, no significant difference was observed, except in the case of two pedestrians. Next, we check the holiday data. In area $w_2$, the difference of the distance tended to be larger. in 2020. This trend is more evident, with the number of pedestrians ranging from two to 13. Similarly, when the number of pedestrians is three or less, the difference of the distance expanded by 0.8 m, and the distance over 2 m is maintained. However, no significant differences were observed in the other observation areas. In summary, the ANN significantly increased after the pandemic, especially in $w_2$ on holidays, which was in front of the southern crosswalk, and this tendency was remarkable when there were few pedestrians in the area. However, no remarkable tendency was observed in other observation areas.

For reference, Figures 20 and 21 show the comparison between the median ANNs of weekdays and holidays for each observed area in 2018 and 2020, respectively. $w_1$ tended to have a larger ANN on weekdays in both years. $w_2$ had the same trend as $w_1$ in 2018, but neither day was predominant in 2020. Unlike $w_1$ and $w_2$, $w_3$ tended to have a larger ANN on holidays, but the median difference was not large.

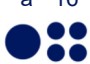

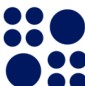

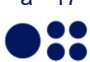

Figure 18. Comparison between ANN medians of 2018 and 2020 on weekdays for each observed area

Figure 19. Comparison between ANN medians of 2018 and 2020 on holidays for each observed area

(a) $w_1$

(b) $w_2$

(c) $w_3$

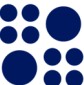

Figure 20. Comparison between ANN medians of weekday and holiday in 2018 for each observed area

Figure 21. Comparison between ANN medians of weekday and holiday in 2020 for each observed area

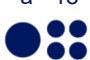

### 8.3 Results of K-function

Figures 22, 23, and 24 show a comparison of the K-function on weekdays. K-function graphs were created for each number of pedestrians; however, because it would be redundant to show all of them, three types of graphs are shown for three (relatively low density), nine (middle density), and 15 (high density) pedestrians in each observation area. $N_{year}$ denotes the number of corresponding images in a year. If the number of pedestrians is missing when calculating the K-function, or $N_{year}$ is less than 10, a graph that is closer to the original number of pedestrians is alternatively shown if available; otherwise, the graph is not shown. Figures 25, 26, and 27 show the graphs for the holidays.

First, we confirmed the results for weekdays. In area $w_1$, except for the case of three pedestrians, the K-function curve in 2018 is larger than that in 2020. Specifically, pedestrians tended to be slightly more concentrated in 2020 under low-density conditions. However, significant differences were observed when $r = 3$–5 m, indicating that physical distance was maintained to some extent. Contrastingly, in medium-density and higher situations, the overall pedestrian concentration tended higher in 2018. In the case of area $w_2$, the relationship between the curves is the opposite of that of $w_1$, and pedestrian concentration tended to be higher in 2020, especially for medium-density and higher. In the case of $w_3$, although comparisons could not be made under high-density conditions, it appears that there was a slight tendency toward centralization in 2018 for low-density conditions.

Next, we confirmed the results for the holidays. First, in area $w_1$, when the number of pedestrians is three or 15, there is a significant tendency for pedestrians to be concentrated in a relatively short range of $r = 2$–3 m or 1–4 m, respectively, in 2018. However, when the number of pedestrians is nine, there is a tendency toward concentration in 2020 in a slightly more distant range of $r = 2.2$–6.5 m. Although the overall trend is not consistent, there is a tendency for the physical distance to be slightly larger in 2020, or for pedestrians to become more concentrated while maintaining a distance of 2 m. In area $w_2$, the results are straightforward: regardless of the number of pedestrians, there is consistently a significant trend toward greater concentration in 2018 in the short-to-medium range of approximately $r = 1$–5 m. Finally, in area $w_3$, the difference between the two years is insignificant.

In summary, the physical distance after the pandemic in Osaka City in 2020 tended to increase more on holidays than on weekdays. This was dependent on the observation area and this tendency was strongly observed in area $w_2$. However, this tendency was weak in area $w_1$, and hardly observed in area $w_3$.

## 9. Discussion

This section discusses two perspectives: the practicality of the methods used, and the interpretation of physical distances.

### 9.1 Practicality of pedestrian detection

Thirty-minute video clips at 1 fps were taken from 12:15 to 16:45 on each recording day. A total of 7,200 images were used for pedestrian detection and their coordinates were projected onto a plan view. The number of pedestrians detected per image was approximately 15, as listed in Table 4, and the cost of manually and visually extracting pedestrians from such a large number of images is impractical; therefore, this research could only be realized by applying object detection. As described in Section 4.2, training only with CrowdHuman dataset, an image dataset dedicated to crowds, did not significantly improve the accuracy compared with YOLO trained with MS COCO, the base dataset. Therefore, additional training with still image samples of the target area significantly improved the detection accuracy. Additional training using target area images is an important and fundamental technique to increase the utility of pedestrian detection. It should be pointed out that the additional training image data were created using only still images from the holidays in 2018; therefore, they did not adequately cover street trees with thicker foliage in 2020, or variations in weather conditions.

The accuracy of the physical distance was evaluated by estimating the error of the projective transformation and the distance between the two participants walking with a 1 m string. First, the error in the projective transformation was 0.17 m on average. However, this error is not considered to be a major problem in this study because knowledge of the relative distance is sufficient rather than the absolute coordinates. However, the accuracy evaluation of the distance between the two participants was performed using a simple measurement method because the location was on a busy public street, and it was difficult to make a

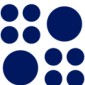

Medians of K-function for 3 pedestrians in w1: 2018_weekday_s and 2020_weekday_s

(a) 3 pedestrians ($N_{2018} = 130, N_{2020} = 518$)

Medians of K-function for 3 pedestrians in w2: 2018_weekday_s and 2020_weekday_s

(a) 3 pedestrians ($N_{2018} = 255, N_{2020} = 771$)

Medians of K-function for 9 pedestrians in w1: 2018_weekday_s and 2020_weekday_s

(b) 9 pedestrians ($N_{2018} = 312, N_{2020} = 176$)

Medians of K-function for 9 pedestrians in w2: 2018_weekday_s and 2020_weekday_s

(b) 9 pedestrians ($N_{2018} = 255, N_{2020} = 63$)

Medians of K-function for 15 pedestrians in w1: 2018_weekday_s and 2020_weekday_s

(c) 15 pedestrians ($N_{2018} = 143, N_{2020} = 11$)

Figure 22. Comparison between K-function medians on weekdays at $w_1$ in 2018 and 2020

Medians of K-function for 13 pedestrians in w2: 2018_weekday_s and 2020_weekday_s

(c) 13 pedestrians ($N_{2018} = 122, N_{2020} = 29$)

Figure 23. Comparison between K-function medians on weekdays at $w_2$ in 2018 and 2020

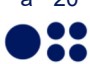

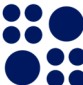

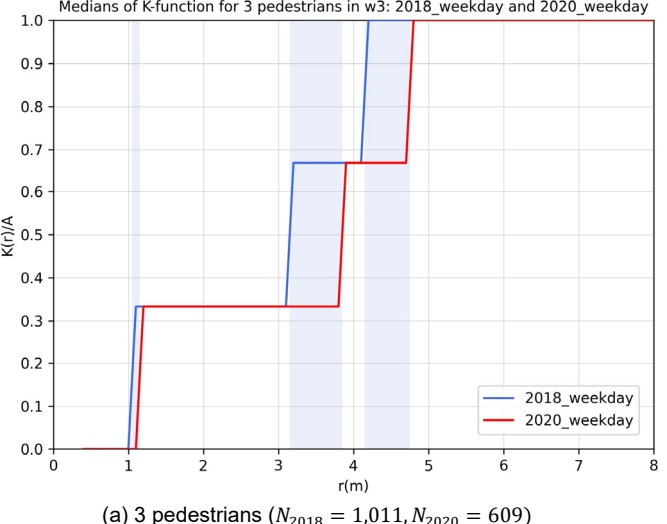

(a) 3 pedestrians ($N_{2018} = 1{,}011, N_{2020} = 609$)

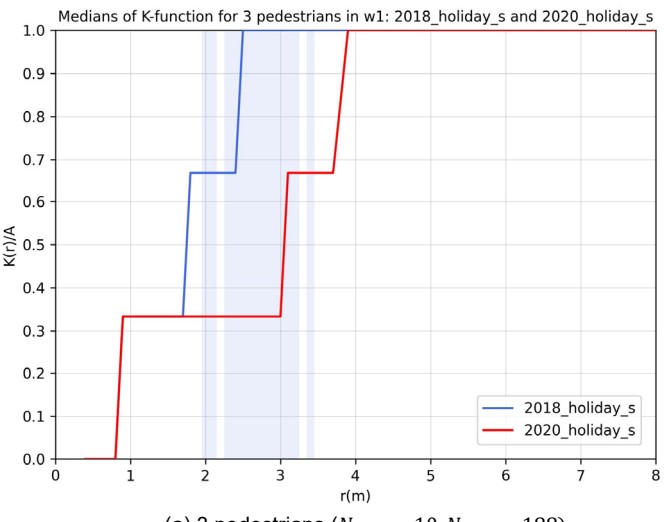

(a) 3 pedestrians ($N_{2018} = 10, N_{2020} = 188$)

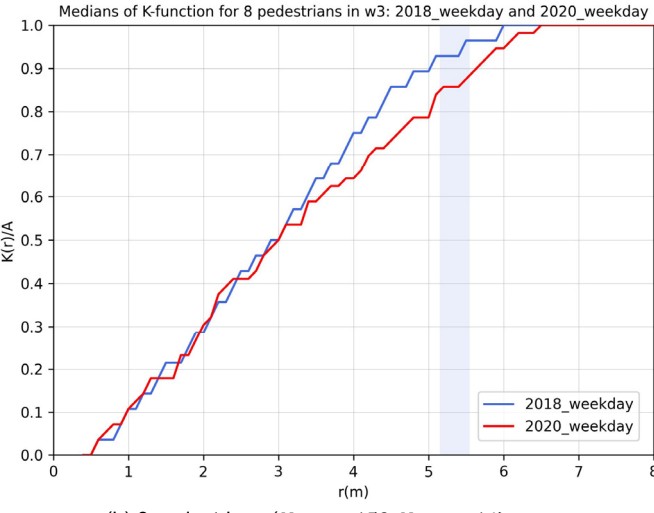

(b) 8 pedestrians ($N_{2018} = 159, N_{2020} = 14$)

Figure 24. Comparison between K-function medians on weekdays at $w_3$ in 2018 and 2020

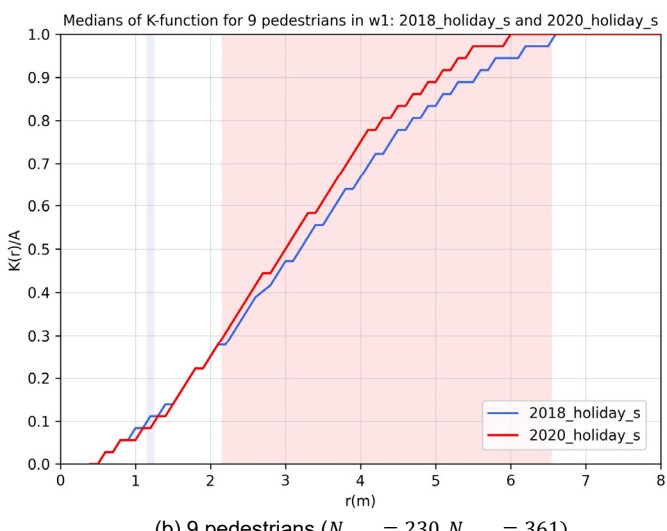

(b) 9 pedestrians ($N_{2018} = 230, N_{2020} = 361$)

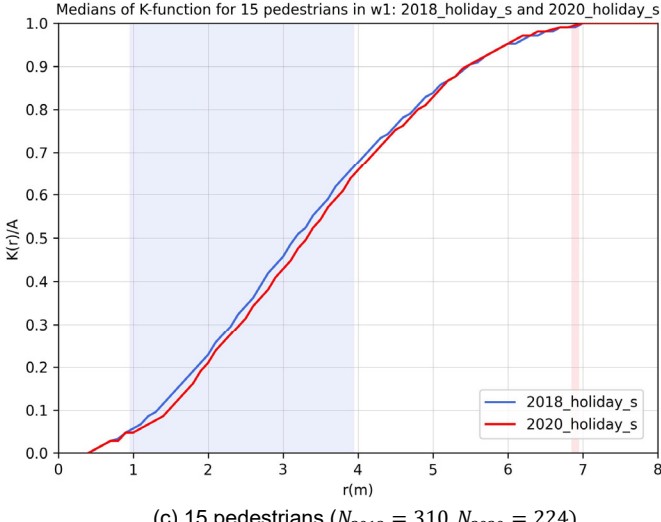

(c) 15 pedestrians ($N_{2018} = 310, N_{2020} = 224$)

Figure 25. Comparison between K-function medians at $w_1$ on holidays in 2018 and 2020

Medians of K-function for 3 pedestrians in w2: 2018_holiday_s and 2020_holiday_s

(a) 3 pedestrians ($N_{2018} = 95, N_{2020} = 531$)

Medians of K-function for 3 pedestrians in w3: 2018_holiday and 2020_holiday

(a) 3 pedestrians ($N_{2018} = 923, N_{2020} = 1014$)

Medians of K-function for 9 pedestrians in w2: 2018_holiday_s and 2020_holiday_s

(b) 9 pedestrians ($N_{2018} = 321, N_{2020} = 225$)

Medians of K-function for 9 pedestrians in w3: 2018_holiday and 2020_holiday

(b) 9 pedestrians ($N_{2018} = 212, N_{2020} = 77$)

Medians of K-function for 15 pedestrians in w2: 2018_holiday_s and 2020_holiday_s

(c) 15 pedestrians ($N_{2018} = 186, N_{2020} = 24$)

Medians of K-function for 13 pedestrians in w3: 2018_holiday and 2020_holiday

(c) 13 pedestrians ($N_{2018} = 50, N_{2020} = 17$)

Figure 26. Comparison of median of K-function between 2018 and 2020 on holiday in $w_2$

Figure 27. Comparison of median of K-function between 2018 and 2020 on holiday in $w_3$

rigorous measurement. Because no restraining device was used to keep the distance constant, the distance probably varied periodically in accordance with their walking rhythm. Furthermore, the distance between the participants was considered to be slightly wider than 1 m because tension was applied to the string when they walked. Thus, the average distance between them was 1.08 m. It is considered sufficient to distinguish physical distances at a level of 0.1–0.2 m at most; therefore, we consider this level of error to be within the acceptable range. In any case, however, it is clear that improving positional accuracy will continue to be important in the future.

## 9.2 Discussion of results

This section discusses the relationship between the two physical distance indices, the difference in physical distance between standing and walking, weekdays and holidays, and $w_1$ and $w_2$.

First, the relationship between the two physical distance indices is discussed. ANN is an index defined by the distance to the nearest pedestrian neighbor, which is not a problem when pedestrians are moving alone; however, when pedestrians are in close groups and their personal space might be narrower, the evaluation of physical distance is biased. Therefore, we attempted to complement physical distance as a distribution over the entire observation area using the K-function method. To understand the relationship between these indicators, Figure 18(b) for the weekday ANN in $w_2$, and Figure 23 for the K-function, were used as examples. In Figure 23, the numbers of pedestrians are three, nine, and 13, which tend to be significantly more concentrated in 2018, 2020, and 2020, respectively. However, by checking the values of the corresponding number of pedestrians in Figure 18(b), the physical distances are significantly larger in 2020 (three pedestrians), significantly larger in 2018 (nine pedestrians), and not significantly different (13 pedestrians). Owing to the inverse relationship between the two indices, there is some correspondence between the cases with three and nine pedestrians. However, when there were 13 pedestrians, the K-function showed a significant difference even when there was no significant difference in the ANN. The K-function is more difficult to understand than the ANN because it is plotted for each number of pedestrians and a radius dimension is added; however, it was demonstrated that the K-function can detect significant differences even when the ANN does not.

Next, we discuss the difference in the physical distance between standing and walking. As stated previously, $w_3$ is basically a space for passing, whereas $w_1$ and $w_2$ are areas before the pedestrian crossings, where standing and walking switch periodically. Therefore, for the sake of contrast, $w_1$ and $w_2$ were analyzed using data only from standing. Compared with the other areas, the difference in the two indicators between the years was less significant at $w_3$. Therefore, the physical distance after a pandemic is likely to be more easily recognized when standing in place than when walking.

We then discuss the difference in the physical distance between weekdays and holidays. Figures 20 and 21 show that the ANN tended to be significantly larger on weekdays than on holidays at $w_1$ and $w_2$. This suggests the obvious that people are more likely to go out with their family and close friends on holidays. However, $w_2$ in 2020 was the only exception, showing no difference according to the day of the week. Furthermore, the K-function also shows a reversal depending on the day of the week. First, in the case of $w_1$, comparing the results of Figures 22 and 25, the weekdays in 2020 tend to be more concentrated in the case of low density, while holidays are more concentrated in the case of medium density. In the case of $w_2$, comparing Figures 23 and 26 on the weekdays pedestrians were more concentrated in 2020 and they were more concentrated on the holidays in 2018. However, the concentration in 2020 starts after 2 m, except for Figure 23(c), suggesting that pedestrians were clustered to some extent while moving away from each other. This is related to the difference in the trends between $w_1$ and $w_2$ in the previous discussion. Table 4 shows that the number of pedestrians was higher in $w_1$, which was located on the north-south axis, and its number was approximately 1.5 times higher than that of $w_2$, which was located on the east-west axis. Furthermore, the number of pedestrians in 2020 was approximately half of that in 2018. It was noticeable from the video that pedestrians in both $w_1$ and $w_2$ were engaged in group activities, such as family and friends, during the 2018 holiday. However, during the 2020 holiday, few pedestrians were observed in groups at $w_2$, although this trend remained unchanged at $w_1$. In this area, the north-south axis along Midosuji Avenue is the main pedestrian flow line for sightseeing and leisure activities, while the east-west axis is not as popular. In 2018, before the pandemic, people were socializing in Osaka's main downtown area, and many pedestrians could be seen outside the north-south axis. However, after the pandemic, walking with close friends is assumed to have decreased owing to self-restraint in going out and other factors, except on the north-south axis. Additionally, there was no clear

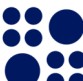

change in physical distance before and after the pandemic in $w_1$ compared with that in $w_2$. In $w_1$, this is likely the result of a combination of the following factors: in 2018, especially in the case of low pedestrian density, the ANN was as high as it was in 2020; the installation of fences, which did not exist in 2018, restricted access to pedestrian crossings, changing the space to a denser pedestrian area; and in the north-south axis, in 2020, there were still some pedestrian activities involving close friends and group activities.

## 10. Concluding remarks

In this study, we used deep learning and statistical techniques to compare and verify the change in physical distance between pedestrians on the same sidewalk in a large Japanese city, before and after the COVID-19 pandemic.

Videos were recorded at the Namba intersection in Osaka City on weekdays and holidays in October 2018 and 2020, from which a large number of still images were extracted. YOLOv3 was trained on the CloudHuman dataset and local images to achieve a high object detection accuracy (mAP) of 0.87, and was used for detecting pedestrians in the images. The average error of the projection transformation of the still image coordinates to plane coordinates with nine correspondence points was approximately 0.17 m. We conducted a physical distance survey in the field and evaluated the accuracy with the help of tracking methods. It was confirmed that the distance distribution by object detection and projection transformation was normally distributed with a mean of 1.08 m and a standard deviation of 0.15 m for an expected distance of 1 m between the participants.

The ANN and K-function were defined as indices of physical distance. Two areas facing the pedestrian crossing, and three observation areas of the same area not facing the pedestrian crossing, were set. Overall, we established that the number of pedestrians in 2020 was approximately half that in 2018, the ANN increased by approximately 0.2–0.3 m after the pandemic, and it was smaller on holidays than on weekdays. Next, a detailed analysis was conducted for each observation area. The main result obtained was that the physical distance significantly increased after the pandemic in $w_2$, facing the crosswalk on the east-west flow line, especially on holidays. More specifically, the ANN expanded significantly over a wide range of 2–13 pedestrians in the area, and the tendency of the K-function to centralize was also significantly higher in 2018. On weekdays, the ANN was also significantly higher when there were up to five pedestrians in the area. However, the changes in $w_1$ and $w_3$ were not as pronounced as those in $w_2$ were. In summary, the overall physical distance tended to be higher in 2020, but in some cases it was larger in 2018, depending on the observation area and the day of the week. These results suggest that physical distance after a pandemic is more likely to be perceived during dwell time than during walking, and that it depends on pedestrian density, relationships among pedestrians, and changes in space, in addition to people's sense of crisis pertaining to COVID-19.

### Acknowledgements

We thank Mr. Takashi Ohira, manager of Namba Midosuji Hall of Kohmei Corporation, Ltd., for providing the location of the video shooting.

### Declaration of competing interests

The authors declare no potential conflicts of interest with respect to the research, authorship, and/or publication of this paper.

### Funding

This research was partially supported by a Grant-in-Aid for Scientific Research (B) (21H01509) and Nikken Sekkei Research Institute.

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

## Appendix

*1 The mAP takes the range [0,1], with values closer to 1 indicating a higher object detection accuracy. The mAP is explained in [40]. Note that YOLO's mAP is calculated with an intersection over union (IOU) threshold of 0.5 by default.

抄録

 新型コロナウィルスは 2020 年以降，世界的な広がりを見せた．世界保健機関では人と人との物理的距離を 1m 以上，日本の場合は 2m 以上を保つことを推奨している．以上の背景から本研究では，新型コロナウィルス流行前後の日本の大都市における歩道上の歩行者間の物理的距離を，動画から深層学習の技術を用いて検出するアプローチによる統計的に比較・検証した．新型コロナウィルス流行前の映像は，2018 年 10 月に大阪市御堂筋の難波地区で撮影されたものである．比較のため，2020 年 10 月に同じ場所で新たに動画を撮影した．物体検出手法として一般的な YOLO v3 SPP を適用し，歩道上の歩行者一人一人を自動的に抽出し，精度評価を行った後，対象エリア内の歩道上に 3 つの観測エリアを設定し，歩行者間の物理的距離を計測した．物理的距離の測定には，平均最近隣距離と K 関数の 2 つの指標を用いた．そして新型コロナウィルス流行前後の歩行者の物理的距離の変化を統計的に比較し，新型コロナウィルス流行後に物理的距離の増加がみられることを確認した．ただし物理的距離の増加は，歩行者の行動状態，密度，対人関係に依存する傾向があることが示唆された．

