# OpenReview forum: "Comparison of Physical Distances Between Pedestrians on a Street in the Central Area of Osaka City Before and After the Covid-19 Pandemic Based on Deep Learning Techniques"
_AIS-J.org/2021/Journal_

### Official Review · AnonReviewer3 · 2022-06-02

**Rating:** 3
**Confidence:** 4

**Review:**

The objective (novelty) of this research is not clearly stated. After pointing the research gap out based on a critical review of previous studies in Chapter 1, the objective of the research should be clearly stated by a sentence such as "the objective of the research …”.

With regard to the above, if the objective of the study is “In this study, we used deep learning and other techniques to statistically compare and verify the change in physical distance between pedestrians on a sidewalk in a large Japanese city, before and after the COVID-19 pandemic.” in the current manuscript, then it looks very weak as an academic paper if it is limited to a comparison survey of the physical distance of people in a particular location in Osaka.
If the above is the objective of this study, it is necessary to consider whether the results of the survey and analysis for a particular location in Osaka obtain findings that can be applied to other locations in Japan, and/or even elsewhere in the world to clarify the generality and limitations of the research fruits.

Or, did the authors propose a method using deep learning to analyze the physical distances of multiple people using image processing to compare fixed points, and then verify/validate this method at a location in Osaka? If yes, the title and the structure of the manuscript need to be reframed.

Next, the contribution that this study will make to the Architectural Informatics Society is not described. The research contribution is usually stated at the end of Chapter 1 or Chapter 2.

"Additionally, at the time of the research, YOLOv3 was used, but as of 2022, YOLOR [37] and other YOLO detectors with higher performance have been proposed; hence, it is necessary to change the object detector to stay updated with the current technology."
If the deep learning network was replaced with a new one, how would the analysis results of this research change? If the results will change, the results in this research might be spoiled, what do the authors think the results of this study would be?

**Overview:**

The topic of the submitted manuscript is suitable for an academic paper for the Architectural Informatics Society. Some improvements as well as comments on unclear points, are listed below.

---

### Official Review · AnonReviewer4 · 2022-06-06

**Rating:** 2
**Confidence:** 3

**Review:**

The paper provides useful insights for readers of the AIS and is appropriate as a journal paper of the AIS. The review of relevant previous studies and the description of experimental and investigative methods in this paper are generally sufficient. However, some of the descriptions are inadequate, and it would be desirable to add or revise the following points.

- P6, L19: There is a description "two participants walked while maintaining spacing with a 1m string. In P9, L17, there is a description "by considering the center of the bottom edge of the bounding box as the position of each pedestrian". However, for a paper that discusses the accuracy of pedestrian location in "cm units," these descriptions are too brief. More details of the method and its validity should be described.

- P14, L2-4: There is a description of the statistics (median) of the ANN results, but it is difficult to read it accurately from Figure 15. Also, Table 5 does not include the median. It is necessary to add a note to either of them.

- P22, P23: Results "consistent" with the hypothesis that the post-pandemic physical distance is wider than the pre-pandemic COVID-19 distance are mentioned in the chapter 9 and 10. However, the opposite result (i.e., the distance between pedestrians is larger before the pandemic) is not mentioned. This case also needs to be analyzed and described.


**Overview:**

The paper discusses the applicability of image processing of movies using deep learning to measure pedestrian locations. After verifying the accuracy of estimating the distance between pedestrians through experiments, the method is applied to movies taken at the same location before and after the COVID-19 pandemic, and the changes in the physical distance between pedestrians during the two periods are evaluated.